

# The urea-creatinine ratio on the seventh day predicts the short-term prognosis of spontaneous intracerebral hemorrhage: a retrospective study

Xingguo Wu[1,*], Ningxiang Qin[1,*], Yiqi Zhang[2], Fahang Yi[1], Xi Peng[2] and Liang Wang[1]

[1] Department of Neurology, The First Affiliated Hospital of Chongqing Medical University, Chongqing, China
[2] Department of Neurology, The Second Affiliated Hospital of Chongqing Medical University, Chongqing, China
[*] These authors contributed equally to this work.

Corresponding author
Liang Wang, wang0128_0@163.com

## ABSTRACT

**Background and Objectives.** This study aimed to investigate the association between hydration status and 90-day functional outcomes in patients with spontaneous intracerebral hemorrhage (SICH).

**Methods.** We conducted a retrospective analysis of 215 SICH patients admitted to the Neurology Department of the First Affiliated Hospital of Chongqing Medical University between January 2021 and September 2023. Demographic characteristics, imaging findings, and laboratory parameters were collected. Patients were stratified into good (modified Rankin Scale [mRS] $\leq$ 2) and poor (mRS > 2) prognosis groups based on 90-day outcomes.

**Results.** Univariate analysis revealed that poor prognosis was associated with advanced age, prolonged hospitalization, and elevated urea nitrogen/creatinine ratio (UCR) on day 7 ($p < 0.05$). Multivariate logistic regression identified three independent predictors of poor outcome: age (OR = 1.059, 95% CI [1.025–1.094], $p < 0.01$), Glasgow Coma Scale score (OR = 0.420, 95% CI [0.308–0.571], $p < 0.01$), and day-7 UCR (OR = 1.095, 95% CI [1.045–1.148], $p < 0.01$). Receiver operating characteristic (ROC) analysis demonstrated that day-7 UCR predicted poor outcomes with an area under the curve (AUC) of 0.72 (95% CI [0.643–0.789]), with an optimal cutoff value of 30.68. Patients with UCR $\leq$ 30.68 had significantly higher rates of favorable outcomes (75.2%) compared to those with UCR > 30.68 (37.9%).

**Conclusion.** Elevated blood UCR (>30.68) on day 7 post-SICH is an independent predictor of unfavorable short-term prognosis.

## INTRODUCTION

Spontaneous intracerebral hemorrhage (SICH), defined as non-traumatic bleeding into the brain parenchyma from rupture of cerebral vessels, such as capillaries, internal cerebral veins, and arteries, represents a critical neurological emergency with substantial

morbidity and mortality (*Greenberg et al., 2022*). Hypertension constitutes the primary etiology, responsible for 70–80% of cases, with an annual incidence of 12–15 individuals per 100,000. The clinical course is often severe, with approximately 75% of patients experiencing acute-phase disability and the overall mortality ranges from 30% to 50% (*Greenberg et al., 2022*; *Sheth, 2022*). Notably, lobar hemorrhages demonstrate distinct pathophysiological characteristics and worse early outcomes compared to deep subcortical hemorrhages, primarily due to their association with cerebral amyloid angiopathy rather than hypertensive vasculopathy (*De Mendiola et al., 2023*).

The role of hydration status in SICH management remains controversial. While some evidence suggests potential benefits of controlled dehydration, including reduced cerebral perfusion pressure, decreased perihematomal edema, and lower intracranial pressure through induced hypovolemia and hypernatremia (*Frey et al., 1994*; *Koenig et al., 2008*; *Diringer et al., 2011*; *Schrock, Glasenapp & Drogell, 2012*), these advantages must be weighed against significant risks. Excessive dehydration may compromise cerebral perfusion during the critical acute phase, exacerbate patient fatigue, impair neurological recovery mechanisms, and increase thromboembolic complications (*Bhalla et al., 2000*; *Kelly et al., 2004*; *Font, Arboix & Krupinski, 2010*; *Acciarresi, Bogousslavsky & Paciaroni, 2014*).

Clinical assessment of dehydration in stroke patients presents unique challenges due to nonspecific symptoms, necessitating reliance on objective biomarkers (*Bahouth, Gottesman & Szanton, 2018*). The blood urea nitrogen-to-creatinine ratio (UCR) has emerged as a particularly valuable indicator, with values $\geq 15$ demonstrating reliable correlation with dehydration status in patients with preserved renal function (*Schrock, Glasenapp & Drogell, 2012*; *Liu et al., 2014*; *Shi et al., 2019*). Elevated UCR has been consistently associated with worse outcomes in stroke populations (*Schrock, Glasenapp & Drogell, 2012*; *Cortés-Vicente et al., 2019*; *Deng et al., 2019*) and serves as an important prognostic marker in other acute conditions (*Rowat, Graham & Dennis, 2012*; *Schrock, Glasenapp & Drogell, 2012*; *Liu et al., 2014*; *Lacey et al., 2019*; *Liu et al., 2019*). However, current evidence regarding hydration status in SICH remains limited, with few studies examining its impact on short-term mortality and none investigating optimal dehydration thresholds (*Lehmann et al., 2021*).

The study aims to elucidate the relationship between the hydration status (quantified by UCR) and 90-day functional outcomes in SICH patients, and establish evidence-based thresholds for dehydration management to guide clinical decision making.

## MATERIALS AND METHODS

### Study population

We conducted a retrospective cohort study of consecutive patients with SICH admitted to the Department of Neurology at the First Affiliated Hospital of Chongqing Medical University between January 2021 and September 2023. The inclusion criteria were as follows: 1. Aged $\geq 18$ years; 2. Primary SICH confirmed by neuroimaging and managed conservatively (*Greenberg et al., 2022*); 3. Hospitalization duration $\geq 7$ days. The exclusion criteria were as follows: 1. Secondary hemorrhage (trauma, vascular malformations, tumors, or aneurysms); 2. Hospital stay <7 days; 3. Extensive cerebral hemorrhage requiring

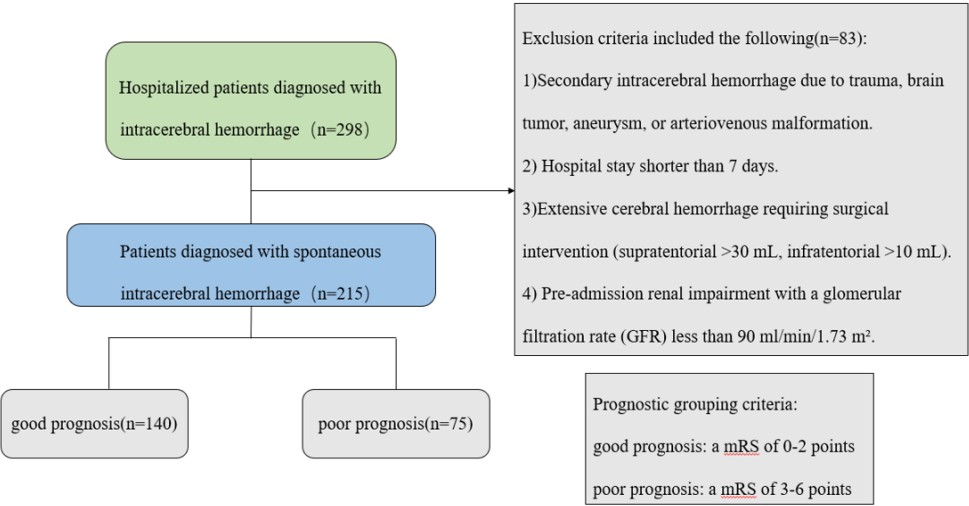

**Figure 1  Patient screening flow chart.**

surgical intervention (supratentorial >30 mL, infratentorial >10 mL); 4. Pre-existing renal impairment with a glomerular filtration rate (GFR) less than 90 ml/min/1.73 m$^2$; 5. Hemorrhagic transformation of ischemic stroke. The study was approved by the Ethics Committee of the First Affiliated Hospital of Chongqing Medical University (K2024-151-01). All participants provided written informed consent before enrollment into this study.

## Clinical assessment

Clinical data collection included demographic characteristics (age, sex), admission Glasgow Coma Scale (GCS) scores (*Mehta et al., 2019*), and detailed hematoma characteristics. Hematoma volume was calculated using the a×b×c/2 method on initial CT imaging (*Kothari et al., 1996*), with documentation of location and presence of intraventricular extension. Comprehensive laboratory tests included urine specific gravity, routine blood tests, electrolyte function tests, coagulation function tests, and liver and kidney function tests. All patients were admitted to the neurology ward, and renal function data were collected on the first and seventh days of hospitalization to calculate the blood UCR.

## Outcome measures

Neurological outcomes were assessed at 90 days post-discharge through either clinic follow-up or structured telephone interviews using the modified Rankin Scale (mRS) (*Sulter, Steen & De Keyser, 1999*). Patients were stratified into good (mRS ≤ 2) and poor (mRS > 2) outcome groups for comparative analysis (Fig. 1).

## Statistical analysis

Statistical analyses were performed using SPSS 25.0 and R 4.3.0. Continuous variables were assessed for normality using the Kolmogorov–Smirnov test, with normally distributed data presented as mean ± standard deviation (analyzed *via* independent t-tests) and non-normal data as median with interquartile range (analyzed *via* Mann–Whitney U tests). Categorical

variables were expressed as frequencies and percentages, with between-group comparisons conducted using $\chi^2$ or Fisher's exact tests as appropriate. Multivariate logistic regression identified independent predictors of outcomes, while receiver operating characteristic (ROC) curve analysis determined optimal cutoff values. A two-tailed $p$-value $<0.05$ was considered statistically significant.

## RESULTS

### Study population and baseline characteristics

A total of 215 patients with SICH were included in this study, all of whom completed 90-day follow-up. Based on mRS scores, 140 patients (65.1%) achieved good functional outcomes (mRS $\leq 2$; 94 males, 46 females), while 75 patients (34.9%) had poor outcomes (mRS $> 2$; 42 males, 33 females). All recorded deaths resulted from non-neurological causes (primarily cardiovascular or infectious complications). A total of 19 patients died, including eight from circulatory failure, seven from pulmonary infection, and four from other causes (*e.g.*, trauma, severe electrolyte disorders).

The demographic and clinical characteristics of the two patient groups were compared, including sex, age, length of hospital stay, GCS score, imaging findings, and laboratory results. The results showed that the age, hospitalization duration, and blood UCR on the 7th day were significantly higher in the poor prognosis group compared to the good prognosis group, with statistically significant differences ($P < 0.05$). The GCS score and potassium ion concentration were lower in the poor prognosis group, also showing statistically significant differences ($P < 0.05$) (Table 1).

### Multivariate regression analysis

Variables demonstrating statistical significance in univariate analysis were incorporated into a multivariate logistic regression model (Table 2). Three independent predictors of poor prognosis emerged: 1. Age: Each additional year increased poor outcome risk by 5.9% (OR = 1.059, 95% CI [1.025–1.094], $P < 0.01$); 2. Admission GCS: Every 1-point increase reduced risk by 58% (OR = 0.420, 95% CI [0.308–0.571], $p < 0.01$); 3. Day-7 UCR: Each unit increase elevated risk by 9.5% (OR = 1.095, 95% CI [1.045–1.148], $p < 0.01$).

### Predictive value of UCR

An ROC curve analysis was conducted to evaluate the predictive effectiveness of day-7 UCR in determining clinical outcomes of SICH patients. The day-7 UCR had an area under the curve (AUC) of 0.72 (95% CI [0.643 $\sim$ 0.789]), sensitivity of 0.48 (95% CI [0.367–0.593]), specificity of 0.843 (95% CI [0.783–0.903]), positive predictive value (PPV) of 0.621, negative predictive value (NPV) of 0.752, and diagnostic accuracy of 0.707, with an optimal cutoff value of 30.68 (Fig. 2).

### Internal validation

The optimal day-7 UCR cutoff value of 30.68 was used to divide 215 patients into two groups: $\leq 30.68$ (157 patients) and $>30.68$ (58 patients). Univariate analysis revealed that patients with a UCR $\leq 30.68$ had a significantly higher proportion of a favorable prognosis

**Table 1 Basic characteristics table of study subjects ($n = 215$).**

| Variable | MRS (0, 1, 2) $n = 140$ | MRS (3, 4, 5, 6) $N = 75$ | P |
|---|---|---|---|
| Demographics | | | |
| Sex M/F | 94/46 | 42/33 | 0.11 |
| Age mean ($\pm$SD) | 61.41 ($\pm$13.21) | 69.9 ($\pm$12.23) | <0.01 |
| Hospitalization time, (IQR) | 14.00 (10.25, 19.95) | 17.00 (12.36, 24.55) | <0.01 |
| GCS, (IQR) | 15 (14, 15) | 13 (11, 14) | <0.01 |
| Imaging features | | | |
| location Supratentorial, $n$ (%) | 128 (91.4) | 73 (97.3) | 0.17 |
| Infratentorial, $n$ (%) | 12 (8.6) | 2 (2.7) | |
| Intraventricular extension yes, $n$ (%) | 118 (84.3) | 58 (77.3) | 0.21 |
| no, $n$ (%) | 22 (15.7) | 17 (22.7) | |
| Hematoma morphology regular,$n$ (%) | 153 (95.0) | 70 (93.3) | 0.58 |
| irregular, $n$ (%) | 7 (5.0) | 5 (6.7) | |
| Bleeding volume, (IQR) | 5.67 (3.00, 12.61) | 7.92 (3.74, 15.86) | 0.12 |
| Laboratory indicators | | | |
| Urine specific gravity, (IQR) | 1.030 (1.022, 1.038) | 1.027 (1.021, 1.035) | 0.27 |
| WBC, (IQR) | 7.59 (5.86, 9.80) | 8.07 (6.71, 10.64) | 0.42 |
| HB, mean ($\pm$SD) | 139.06 ($\pm$17.78) | 134.91 ($\pm$16.65) | 0.97 |
| PLT, (IQR) | 197.0 (164.0, 238.5) | 202.0 (155.3, 227.8) | 0.44 |
| NLR (IQR) | 4.07 (2.64, 7.61) | 4.91 (2.13, 8.49) | 0.88 |
| PT (IQR) | 11.3 (10.8, 11.9) | 11.3 (10.6, 12.1) | 0.70 |
| APTT (IQR) | 26.0 (24.7, 27.7) | 25.6 (24.4, 27.8) | 0.39 |
| FIB (IQR) | 3.01 (2.48, 3.72) | 3.04 (2.55, 3.70) | 0.86 |
| ALB (IQR) | 42.0 (39.5, 45.0) | 41.0 (38.8, 45) | 0.08 |
| ALT (IQR) | 22.0 (16.0, 32.0) | 23.0 (16.8, 27.3) | 0.81 |
| AST (IQR) | 25.0 (20.0, 31.0) | 26.0 (22.8, 31.3) | 0.28 |
| Ca (IQR) | 2.25 (2.18, 2.34) | 2.22 (2.14, 2.29) | 0.06 |
| K (IQR) | 3.8 (3.6, 4.1) | 3.7 (3.5, 4.0) | 0.02 |
| Na (IQR) | 140 (138, 141) | 139 (137, 142) | 0.56 |
| BCR d1, (IQR) | 18.84 (15.06, 24.9) | 19.50 (14.94, 25.42) | 0.59 |
| BCR d7, mean ($\pm$SD) | 22.85 ($\pm$7.86) | 30.77 ($\pm$10.03) | <0.01 |

**Table 2 Multivariate logistic regression analysis of short-term prognostic risk factors in patients with spontaneous intracerebral hemorrhage.**

| Variables | Regression coefficient | Standard error | Wald $X^2$ | OR | (95% confidence interval) | P |
|---|---|---|---|---|---|---|
| Age | 0.057 | 0.017 | 11.770 | 1.059 | 1.025, 1.094 | 0.001 |
| Hospitalization time | 0.026 | 0.014 | 3.645 | 1.026 | 0.999, 1.054 | 0.056 |
| GCS | −0.868 | 0.157 | 30.420 | 0.420 | 0.308, 0.571 | 0.000 |
| K | −0.972 | 0.509 | 3.643 | 0.378 | 0.140, 1.026 | 0.056 |
| BCR d7 | 0.091 | 0.024 | 14.475 | 1.095 | 1.045, 1.148 | 0.000 |

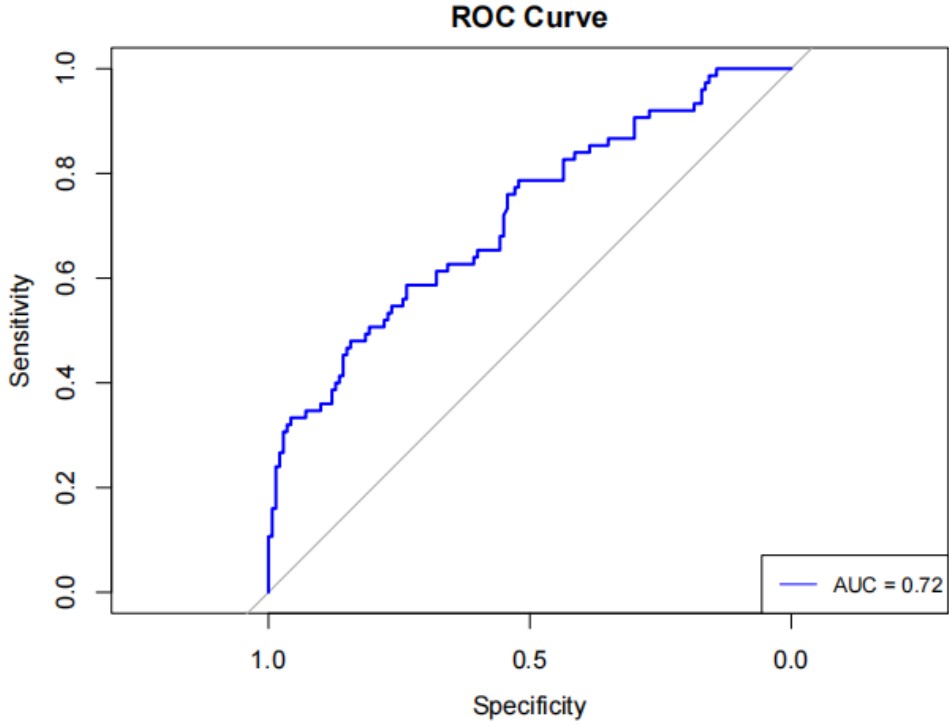

**Figure 2 Serum urea nitrogen creatinine ratio for predicting the prognosis of spontaneous intracerebral hemorrhage ROC curve with AUC labeling.**

(75.2%) than those with a UCR > 30.68 (37.9%), with a statistically significant difference ($P < 0.01$) (Table 3).

### External validation

External validation of the optimal cutoff value of 30.68 for the blood UCR was conducted at the Second Affiliated Hospital of Chongqing Medical University during the same study period. Sixty-four patients with SICH who met the inclusion criteria were included in the validation. There were 56 patients in the group UCR ≤ 30.68 and eight patients in the group UCR > 30.68. Single-factor analysis revealed that the favorable prognosis rate in the group UCR ≤ 30.68 (75.0%) was significantly higher than that in the group UCR > 30.68 (0.0%), with a statistically significant difference ($P < 0.01$) (Table 4).

### DISCUSSION

Our study demonstrates that a blood UCR > 30.68 on day seven post-SICH independently predicts poor 90-day functional outcomes (mRS > 2). This finding carries significant clinical relevance, as excessive dehydration indicated by elevated UCR, may compromise cerebral perfusion through multiple mechanisms: plasma volume contraction, reduced cardiac output, and impaired collateral circulation (*González-Alonso et al., 1995*).

The clinical detection of dehydration presents notable challenges, particularly in patients with SICH. Traditional physical signs such as dry mucous membranes, decreased skin

**Table 3 The impact of the optimal cut-off value (30.68) of urea nitrogen creatinine ratio on short-term prognosis ($n = 215$).**

| Variable | BCR ≤ 30.68 $n = 157$ | BCR > 30.68 $n = 58$ | P |
|---|---|---|---|
| Demographics | | | |
|   Sex M/F | 110/47 | 26/32 | 0.01 |
|   Age mean (±SD) | 62.69 (±13.56) | 68.98 (±12.21) | 0.02 |
| Hospitalization time, (IQR) | 14.45 (10.86, 21.00) | 14.48 (11.55, 22.73) | 0.45 |
| GCS, (IQR) | 15 (14, 15) | 14 (13, 15) | 0.01 |
| MRS 0, 1, 2 (%) | 118 (75.2) | 22 (37.9) | <0.01 |
|    3, 4, 5, 6 (%) | 39 (24.8) | 36 (62.1) | |
| Imaging features | | | |
|   location Supratentorial, $n$ (%) | 144 (71.6) | 57 (28.4) | 0.16 |
|     Infratentorial, $n$ (%) | 13 (92.9) | 1 (7.1) | |
|   Intraventricular extension yes, $n$ (%) | 125 (71.0) | 51 (29.0) | 0.16 |
|     no, $n$ (%) | 32 (82.1) | 7 (17.9) | |
|   Hematoma morphology regular, $n$ (%) | 148 (72.9) | 55 (27.1) | 1.00 |
|     irregular, $n$ (%) | 9 (75.0) | 3 (25.0) | |
|   Bleeding volume, (IQR) | 6.93 (3.43, 13.11) | 5.56 (2.32, 14.04) | 0.28 |
| Laboratory indicators | | | |
|   Urine specific gravity, (IQR) | 1.029 (1.022, 1.038) | 1.028 (1.021, 1.032) | 0.74 |
|   WBC, (IQR) | 7.75 (6.02, 10.27) | 8.01 (5.96, 9.55) | 0.70 |
|   HB, (IQR) | 139.0 (130.0, 149.0) | 132.5 (124.8, 145.8) | 0.11 |
|   PLT, (IQR) | 199.0 (165.0, 236.0) | 190.0 (152.5, 227.0) | 0.56 |
|   NLR (IQR) | 4.11 (2.63, 8.14) | 4.18 (2.43, 7.76) | 0.89 |
|   PT (IQR) | 11.30 (10.70, 11.90) | 11.25 (10.70, 12.00) | 0.80 |
|   APTT (IQR) | 26.00 (24.75, 28.00) | 25.20 (24.15, 27.02) | 0.32 |
|   FIB (IQR) | 3.03 (2.54, 3.72) | 2.96 (2.42.3.70) | 0.91 |
|   ALB (IQR) | 42.00 (39.00, 45.00) | 41.50 (39.00, 45.00) | 0.48 |
|   ALT (IQR) | 22.00 (16.50, 31.50) | 23.50 (16.00, 27.25) | 0.33 |
|   AST (IQR) | 24.00 (20.00, 30.00) | 29.00 (24.00, 32.25) | 0.01 |
|   Ca (IQR) | 2.25 (2.19, 2.32) | 2.20 (2.13, 2.30) | 0.10 |
|   K (IQR) | 3.8 (3.6, 4.0) | 3.9 (3.6, 4.0) | 0.93 |
|   Na (IQR) | 140 (138, 142) | 139 (137, 141) | 0.23 |

turgor, and subjective thirst may prove unreliable, as they are often influenced by patient perception and clinical context. This diagnostic difficulty is further compounded in elderly patients, who frequently exhibit both diminished thirst perception and age-related declines in renal concentrating ability, making them particularly vulnerable to inadequate hydration. In clinical practice, objective biomarkers including the blood UCR (*Schrock, Glasenapp & Drogell, 2012*; *Crary et al., 2013*; *Liu et al., 2014*) and plasma osmolarity (*Bhalla et al., 2000*) have emerged as valuable tools for assessing hydration status. The UCR offers particular advantages in stroke populations, as serum urea and creatinine measurements are routinely obtained, cost-effective, and provide rapid results. Our study specifically highlights the clinical value of day-7 UCR measurements, which demonstrate superior

**Table 4** External validation of the impact of the 7th day urea nitrogen creatinine ratio on the 90 day prognosis ($n = 64$).

| Variable | BCR ≤ 30.68 $n = 56$ | BCR > 30.68 $n = 8$ | P |
|---|---|---|---|
| Demographics | | | |
| Sex M/F | 41/15 | 5/3 | 0.83 |
| Age, mean (±SD) | 62.64 (13.57) | 74.00 (10.029) | 0.03 |
| Hospitalization time, (IQR) | 11 (9, 14) | 16 (10, 25) | 0.02 |
| GCS, (IQR) | 15 (15, 15) | 15 (11, 15) | <0.01 |
| MRS 0, 1, 2 (%) | 42 (75.0) | 0 (0) | <0.01 |
| 3, 4, 5, 6 (%) | 14 (25.0) | 8 (100) | |
| Imaging features | | | |
| location Supratentorial, $n$ (%) | 52 (92.9) | 6 (75.0) | 0.33 |
| Infratentorial, $n$ (%) | 4 (7.1) | 2 (25.0) | |
| Intraventricular extension yes, $n$ (%) | 39 (69.6) | 4 (50.0) | 0.48 |
| no, $n$ (%) | 17 (30.4) | 4 (50.0) | |
| Hematoma morphology regular, $n$ (%) | 53 (94.6) | 6 (75.0) | 0.50 |
| irregular, $n$ (%) | 3 (5.4) | 2 (25.0) | |
| Bleeding volume, (IQR) | 7.80 (3.38, 11.88) | 7.61 (1.17, 9.92) | 0.97 |
| Laboratory indicators | | | |
| Urine specific gravity, (IQR) | 1.015 (1.015, 1.020) | 1.020 (1.015, 020) | 0.81 |
| WBC, (IQR) | 7.21 (6.15, 9.39) | 7.55 (6.28, 10.60) | 0.63 |
| HB, mean (±SD) | 140.41 (±15.79) | 124.50 (±18.19) | 0.01 |
| PLT, mean (±SD) | 209.64 (±64.34) | 201.25 (±67.04) | 0.73 |
| NLR (IQR) | 4.88 (2.85, 6.92) | 5.22 (3.86, 5.95) | 0.92 |
| PT (IQR) | 12.90 (12.30, 13.40) | 13.70 (13.00, 14.40) | 0.02 |
| APTT (IQR) | 33.70 (32.20, 37.50) | 36.70 (31.30, 40.40) | 0.29 |
| FIB (IQR) | 3.12 (2.79, 3.78) | 3.98 (3.78, 5.80) | 0.02 |
| ALB, mean (±SD) | 42.55 (±3.88) | 39.34 (±6.05) | 0.18 |
| ALT (IQR) | 19.00 (14.00, 31.00) | 15.00 (13.00, 20.00) | 0.38 |
| AST (IQR) | 21.00 (18.00, 24.00) | 22.00 (17.00, 25.00) | 0.51 |
| Ca, mean (±SD) | 2.32 (±0.12) | 2.24 (±0.14) | 0.09 |
| K, (IQR) | 3.61 (3.41, 3.85) | 3.99 (3.37, 4.21) | 0.48 |
| Na, (IQR) | 137.80 (136.40, 140.90) | 137.11 (133.80, 137.90) | 0.10 |

reliability compared to admission values. By this timepoint, patients have typically stabilized from acute resuscitation, and confounding factors such as pre-hospital dietary intake, stress responses, and medication effects are minimized. This allows for a more accurate reflection of true hydration status under standardized hospital care. However, several practical limitations of UCR monitoring warrant consideration. Serial blood sampling may cause patient discomfort, while the inherent delay in laboratory processing could potentially delay therapeutic adjustments. These limitations highlight the need for technological advances in real-time hydration monitoring to better guide fluid management decisions. Despite these challenges, the UCR remains a pragmatic and clinically relevant biomarker for hydration

assessment in stroke patients, supported by its widespread use in existing literature and consistent performance in prognostic studies (*Bahouth, Gottesman & Szanton, 2018*).

Our univariate analysis identified three clinically significant prognostic indicators: advanced age, lower GCS scores, and elevated day-7 UCR. These factors demonstrated strong associations with poorer 90-day functional outcomes, suggesting their potential utility in clinical risk stratification. The relationship between advanced age and worse outcomes likely reflects the cumulative impact of age-related physiological decline. Elderly patients typically present with multiple comorbidities, including renal impairment and reduced physiological reserve, which may compromise recovery potential (*Fried et al., 2001*; *Radholm et al., 2015*). Neurological status at admission, as measured by GCS, proved to be another powerful prognostic indicator. The GCS provides a standardized assessment of consciousness through evaluation of eye opening, verbal response, and motor function (scores 3–15), with lower scores indicating more severe neurological impairment (*Mehta et al., 2019*). On the 7th day post-admission, the UCR effectively reflects the patient's hydration status following standardized treatment protocols. Patients in the adverse prognosis group exhibited higher UCR levels compared to those with favorable prognoses, suggesting that excessive dehydration is not conducive to favorable outcomes in cerebral hemorrhage patients. This finding represents a significant and novel discovery of this study. As stated above, patients with spontaneous intracerebral hemorrhage often present with advanced age and poor consciousness. This group of patients requires objective indicators for the dynamic follow-up of dehydration treatment after admission, rather than continuous dehydration treatment during the acute phase. For such patients, if dehydration treatment is necessary due to significant mass effect after admission, UCR monitoring should be carried out on the seventh day. If the value is >30.68, adjusting the dehydration treatment plan may be safer for the patient's prognosis.

Dehydration—a recognized care quality indicator—was unassessed. It may impact outcomes *via* physiological (*e.g.*, prerenal injury) or care-related (*e.g.*, fluid management) pathways, especially for post-first-week non-neurological deaths. This study represents the investigation into the prognostic value of blood UCR for 90-day outcomes in SICH patients, with external validation enhancing the credibility of our findings. The identification of UCR as a potential biomarker for guiding dehydration therapy endpoints constitutes a significant contribution to clinical practice. However, several important limitations must be acknowledged. The relatively small sample size and exclusion of patients with severe hemorrhage or those requiring surgical intervention may affect the generalizability of our results. Notably, we did not assess intraventricular extension (IVE) of hematoma (*Arboix et al., 2007*), a well-established poor prognostic factor that could confound the observed association between UCR and outcomes. We also did not assess the process of care, like late hospital arrival, ≥24 h in the Emergency Department and Stroke Unit admission (*Fernandes et al., 2022*), which have strong impact on mortality of SICH. Other unmeasured confounders including comorbidities (*e.g.*, infections, dysphagia), alternative causes of elevated UCR (*e.g.*, renal insufficiency, medication effects), and direct measures of hydration status limit our ability to definitively establish dehydration as the primary driver of poor outcomes. The 90-day follow-up period precludes evaluation of long-term

prognosis, while the absence of data on specific dehydration management interventions represents another important knowledge gap. Future multicenter studies incorporating detailed neuroimaging for IVE assessment, comprehensive comorbidity documentation, direct hydration metrics, and standardized treatment protocols will be essential to validate and extend our findings, ultimately improving risk stratification and fluid management in SICH patients.

## CONCLUSIONS

Our study demonstrates that a blood UCR exceeding 30.68 on day seven post-admission is a significant predictor of poor 90-day outcome in SICH patients, suggesting its potential utility in guiding dehydration therapy. While these findings provide a promising biomarker for clinical decision-making, further large-scale validation studies are needed to confirm the optimal UCR threshold and establish evidence-based protocols for its application in routine patient management. This research direction could lead to more precise and individualized treatment strategies for SICH patients.

### Funding
This work was funded by High-Level Medical Reserved Personnel Training Project of Chongqing (2020GDRC019). The funders had no role in study design, data collection and analysis, decision to publish, or preparation of the manuscript.

### Grant Disclosures
The following grant information was disclosed by the authors:
High-Level Medical Reserved Personnel Training Project of Chongqing: 2020GDRC019.

### Competing Interests
The authors declare there are no competing interests.

### Author Contributions
- Xingguo Wu conceived and designed the experiments, performed the experiments, analyzed the data, prepared figures and/or tables, and approved the final draft.
- Ningxiang Qin performed the experiments, prepared figures and/or tables, and approved the final draft.
- Yiqi Zhang analyzed the data, prepared figures and/or tables, and approved the final draft.
- Fahang Yi analyzed the data, prepared figures and/or tables, and approved the final draft.
- Xi Peng conceived and designed the experiments, authored or reviewed drafts of the article, and approved the final draft.
- Liang Wang conceived and designed the experiments, authored or reviewed drafts of the article, and approved the final draft.

## Human Ethics

The following information was supplied relating to ethical approvals (i.e., approving body and any reference numbers):

The Ethics Committee of the First Affiliated Hospital of Chongqing Medical University approved this research (K2024-151-01).

## Data Availability

Raw data is available in the Supplemental Files.

## Supplemental Information

Supplemental information for this article can be found online at http://dx.doi.org/10.7717/peerj.19874#supplemental-information.

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
