# Peer review of "The urea-creatinine ratio on the seventh day predicts the short-term prognosis of spontaneous intracerebral hemorrhage: a retrospective study"

_PeerJ, doi:10.7717/peerj.19874_

## Round 0.1 · original submission · Major Revisions

·

Basic reporting

The authors present the results of a single-center retrospective clinical study aimed to analyze the association between hydration status and short-term prognosis (90 days) in patients with spontaneous intracerebral hemorrhage (SICH). A total of 215 patients were included in the study. The authors found that 7th-day blood urea/creatinine ratio (UCR) (OR = 1.095) -as a dehydration marker- was an independent predictor of poor outcome in SICH. Specifically, a blood UCR ratio >30.68 on day 7 after SICH was associated with poor prognosis at 90 days (mRS >2). The paper is potentially interesting, but some aspects of the manuscript may be improved taking into account the following points

Experimental design

The Methods are described with suficient information to replicate.

Validity of the findings

-It is mandatory to describe the causes of death (neurological and non-neurological) in the study sample.

-In the Results, it would be interesting to add the sensitivity, specificity, positive predictive value, negative predictive value, and diagnostic accuracy in the multivariate analysis.

Additional comments

-Please describe all acronyms used in the Abstract (ex: “UCR”).

-Typographic errors in the text (line 154) should be corrected (However, However).

-The authors should mention in the Introduction section that acute lobar cerebral hemorrhages present a different clinical profile and a more severe early prognosis than deep subcortical intracerebral hemorrhages (Biomedicines 2023 Jan 16;11(1):223) and should add that non-hypertensive mechanisms of intracerebral hemorrhages predominate in the lobar location.

-It would be interesting to add in the Discussion section a comment on the relevance of intraventricular extension of cerebral hematoma as a determinant of poor early outcome in humans (BMC Neurol. 2007 Oct 5;7:32. doi: 10.1186/1471-2377-7-32. PMID: 17919332; PMCID: PMC2169250). Add and comment on the reference. Did the authors consider this in their study?

-The opinion of the authors on future lines of research on this topic should be added in the text.

8. Please check reference #11

Reviewer 2 ·

Basic reporting

The manuscript is clear, unambiguous and easy to follow. In general the references are updated and allow proper contextualization of the study.

Experimental design

Wu et al. present an interesting study aiming to explore whether the seventh-day urea-creatinine ratio (UCR) can predict the prognosis of intracerebral hemorrhage (ICH). The study merits scientific attention as it points to a simple and potentially useful tool to help signal the risk of poor outcomes. However, a key limitation is that the authors did not account for several other comorbidities or complications that may be associated with both dehydration and outcomes—such as infections and severe dysphagia.

Validity of the findings

Comments and Suggestions:

1.In the abstract and manuscript, for the variable Glasgow Coma Scale score (OR = 0.420, 95% CI: 0.308–0.571, p < 0.01), please clarify the unit of analysis—i.e., is this per point on the scale?

2.Similarly, for 7th-day UCR (OR = 1.095, 95% CI: 1.045–1.148, p < 0.01), please specify the unit used for the UCR (e.g., per unit increase?).

3.The authors should discuss the potential risk of bias, particularly due to confounding factors. Many causes of dehydration may themselves be responsible for poor outcomes, rather than dehydration acting as an independent predictor.

4.Please correct the patient screening flowchart. 2) Hospital stay shorter than 7 days

5.The sentence "A total of 215 patients with SICH were enrolled in this study" should be revised to "A total of 215 patients with SICH were included in this study", as "enrolled" suggests a prospective study design, which may not be appropriate here.

5.Dehydration is recognized as an indicator of the quality of care processes (https://doi.org/10.1016/j.jocn.2022.05.021), which was not addressed in this study. The authors should acknowledge this limitation—especially because the study focuses on outcomes such as non-neurological deaths occurring after the first week.

6.Related to point 5, the authors should also acknowledge the limitation of not reporting how dehydration was managed once identified, as this could significantly influence patient outcomes

---

## Round 0.2 · accepted · Accept

Thanks for addressing the reviewers' comments and congratulations.

·

Basic reporting

The authors present the results of a nice retrospective single-center clinical analysis identifying a predictive model in which blood urea/creatinine ratio, as a dehydration marker, is an independent predictor of poor outcome for symptomatic intracerebral hemorrhage. The topic presented in the paper is controversial and poorly studied, but clinically interesting.

Experimental design

The research question is well defined, and the methods are described with sufficient detail.

Validity of the findings

Conclusions are linked to the original research question.

Additional comments

I thank the authors for addressing the issues in my initial review. I am satisfied with the additional sentences added to the manuscript and have no additional suggestions to make

Reviewer 2 ·

Basic reporting

The authors have responded to all my comments. I do not have any further points to address.

Experimental design

-

Validity of the findings

-